# Improved chemotherapy modeling with RAG-based immune deficient mice

Mark Wunderlich[1]*, Nicole Manning[1], Christina Sexton[1], Anthony Sabulski[2], Luke Byerly[2], Eric O'Brien[2], John P. Perentesis[2], Benjamin Mizukawa[2], James C. Mulloy[1]*

**1** Division of Experimental Hematology and Cancer Biology, Cancer and Blood Disease Institute, Cincinnati Children's Hospital Medical Center, Cincinnati, Ohio, United States of America, **2** Division of Hematology and Oncology, Cancer and Blood Disease Institute, Cincinnati Children's Hospital Medical Center, Cincinnati, Ohio, United States of America

* mark.wunderlich@cchmc.org (MW); james.mulloy@cchmc.org (JM)

**Data Availability Statement:** All relevant data are within the manuscript and its Supporting Information files.

## Abstract

We have previously characterized an acute myeloid leukemia (AML) chemotherapy model for SCID-based immune deficient mice (NSG and NSGS), consisting of 5 days of cytarabine (AraC) and 3 days of anthracycline (doxorubicin), to simulate the standard 7+3 chemotherapy regimen many AML patients receive. While this model remains tractable, there are several limitations, presumably due to the constitutional Pkrdc$^{scid}$ (SCID, severe combined immune deficiency) mutation which affects DNA repair in all tissues of the mouse. These include the inability to combine preconditioning with subsequent chemotherapy, the inability to repeat chemotherapy cycles, and the increased sensitivity of the host hematopoietic cells to genotoxic stress. Here we attempt to address these drawbacks through the use of alternative strains with RAG-based immune deficiency (NRG and NRGS). We find that RAG-based mice tolerate a busulfan preconditioning regimen in combination with either AML or 4-drug acute lymphoid leukemia (ALL) chemotherapy, expanding the number of samples that can be studied. RAG-based mice also tolerate multiple cycles of therapy, thereby allowing for more aggressive, realistic modeling. Furthermore, standard AML therapy in RAG mice was 3.8-fold more specific for AML cells, relative to SCID mice, demonstrating an improved therapeutic window for genotoxic agents. We conclude that RAG-based mice should be the new standard for preclinical evaluation of therapeutic strategies involving genotoxic agents.

## Introduction

Currently, NOD/SCID IL2Rγ$^{-/-}$ (NSG) mice are the most commonly used strain for engraftment of both normal and malignant human hematopoietic tissues. These mice represent a dramatic improvement over older strains for engraftment of normal HSCs [1] as well as AML and ALL cell lines and patient samples [2]. NSG mice with transgenic expression of human SCF/GM-CSF/IL-3 cytokines (NSGS) further improved AML engraftment efficiency, latency, and levels [3, 4]. Similarly, NRGS mice (NOD/RAG IL2Rγ$^{-/-}$(NRG) mice harboring the same SCF/

**Funding:** This work was funded by an NIH/NCI R50 award (#CA21140, MW), an NIH/NCI R01 award (#CA215504, JCM) and a Cincinnati Children's Hospital ARC award (JCM). The funders had no role in study design, data collection and analysis, decision to publish, or preparation of the manuscript. There was no additional external funding received for this study.

**Competing interests:** The authors have declared that no competing interests exist.

GM-CSF/IL-3 transgene) also exhibited improved engraftment of patient AML samples when compared to NRG [5].

We previously characterized the therapeutic response of AML samples to combined Ara-C and doxorubicin in NSGS mice [6]. Importantly, this model revealed differential response of patient samples to a 5+3 regimen; de novo samples showed delayed disease progression while relapse/refractory samples were resistant. This is consistent with the finding of excellent concordance between the response of a large, diverse group of patient derived xenograft (PDX) models to patient outcome using a variety of therapies [7].

While several groups have successfully employed SCID-based immune deficient mice for studies involving PDX response to standard chemotherapies [8–14], there are limitations for doses, frequency, and prior conditioning. These shortcomings are presumably related to the $Prkdc^{scid}$ mutation, which is responsible for defects in DNA repair [15] and extreme radio-sensitivity [16]. For unknown reasons, these issues are even more pronounced in $IL-2ry^{-/-}$ mice [1]. In contrast, NRG mice tolerate much higher doses of radiation [17] yet retain the ability to engraft human HSCs and give rise to human blood cell levels and subpopulations that are very similar to NSG mice [18]. It is important to recognize that SCID mutation has functional consequences for every cell in SCID mice, while RAG knockout should only affect differentiation and maturation of lymphocytes. Concerns about SCID-related toxicity are not limited to the hematopoietic compartment for PDX models. For example, it is well established that anthracyclines, which are a common agent in leukemia therapy, have significant toxic effects on cardiac tissues which could be exacerbated in the presence of a SCID mutation [19].

One limitation with previous SCID chemotherapy models was the inability to administer repeated cycles of chemotherapy. Current guidelines for adult and pediatric AML call for two induction cycles, followed by additional intensification/consolidation cycles [20, 21]. Repeated cycles in PDX models may allow for more realistic modeling of response and improved efficacy. Another limitation with the SCID-based model is the inability to give chemotherapy after prior conditioning with either gamma irradiation or busulfan injection. Such conditioning is required for reliably robust engraftment of some PDX samples.

In our previous study, we were careful to examine the effects of chemotherapy on both AML and non-malignant host BM cells [6]. We showed increased sensitivity of AML cells to chemotherapy, particularly with doxorubicin. Ara-C had only minimal selective effects on AML, but increased treatment toxicity. However, these experiments were done in SCID mice, which are likely artificially sensitive to DNA damage-inducing chemotherapy. This sensitivity may artificially lower the relative AML response readout. The maximum tolerable doses of chemotherapies are also likely artificially low and sub-optimal for therapeutic effect.

Recent PDX ALL therapy models in NSG mice utilized a 3-drug induction regimen with vincristine, dexamethasone and L-asparaginase (VXL). This approach has been successfully used along with bioluminescent imaging [22] and combined with Bcl inhibitors [23, 24]. A 4-drug induction protocol (VXL+daunorubicin) optimized for T-ALL engrafted NOD/SCID resulted in 2 of 4 PDX developing signs of resistance [25]. We are unaware of a 4-drug induction protocol for NSG mice. One likely pitfall is increased sensitivity of NSG to anthracyclines [1].

Here, we determined the sensitivity of RAG-based mice (NRG and NRGS) to standard AML and 4-drug ALL induction chemotherapy. RAG-based mice tolerated significantly higher doses of daunorubicin/Ara-C, repeated cycles of therapy as well as combination with busulfan conditioning. Interestingly, we also uncovered a differential activity of doxorubicin and daunorubicin in RAG mice that highlights the importance of full characterization of therapeutics in the various immune deficient models. Finally, we showed that RAG-based host BM cells are more resistant to DA therapy, resulting in an approximate 3.8-fold increase in

therapeutic window relative to SCID-based mice. These experiments illustrate the degree to which the choice of host strain may affect results with genotoxic therapies in PDX systems.

## Materials and methods

### Mice

NOD.Cg-Prkdc$^{scid}$Il2rg$^{tm1Wjl}$/SzJ (NSG) and NOD.Cg-Rag1$^{tm1Mom}$Il2rg$^{tm1Wjl}$/SzJ (NRG) mice were obtained from Jackson Laboratories. Generation of NOD.Cg-Prkdc$^{scid}$Il2rg$^{tm1Wjl}$Tg (CMV-IL3,CSF2,KITLG)1Eav/MloySzJ (NSGS) [3] and NOD.Cg-Rag1$^{tm1Mom}$Il2rg$^{tm1Wjl}$Tg (CMV-IL3,CSF2,KITLG)1Eav/J (NRGS) [26] have been previously described. All strains were housed and bred in a pathogen-free facility at Cincinnati Children's Hospital in accordance with an IACUC protocol. Veternary Services of Cincinnati Children's Hosptial provided hands on and classroom training concerning proper animal handling for all research staff. Mice (both males and females, aged 8–12 weeks) subjected to chemotherapy protocols were monitored twice daily for signs of toxicity. Mice showing poor mobility, labored breathing, or cumulative weight loss of 30% of their initial body weight were immediately euthanized. These humane endpoints discriminate mice with lethal toxicities from those showing less severe, transient signs of illness from chemotherapy exposure (scruffy appearance and slight hunched posture). Chemotherapy exposed mice were provided moistened food to allow easier feeding and aid hydration. Leukemic mice often rapidly develop hind limb paralysis when tumor burden is high therefore mice with signs of hind limb weakness were also euthanized. Additionally, BM and PB samples were periodically taken from leukemic mice in order to ascertain the level of leukemic burden and to better predict the onset of illness. Bone marrow aspirates were taken from live mice under general anesthesia with isoflurane as previously described in detail [27]. Mice received buprenorphine hydrochloride injections to minimize pain and discomfort before the procedure and after, as necessary. Death was not used as an endpoint for any experiment, however, occasionally mice were found dead, presumably due to rapid progression and onset of disease symptoms and/or toxicities during the overnight hours. This was limited to fewer than 5% of mice involved in our studies. All leukemic and chemotherapy protocols and humane endpoints were reviewed and approved by the Cincinnati Children's Hospital IACUC prior to study initiation.

### Cells

The MA9.3RAS cell line was generated by sequential retroviral expression of MLL/AF9 and NRasG12D cDNAs into umbilical cord blood (UCB) CD34+ cells, as described previously [3, 28], and was maintained in IMDM/20%FBS. 2X10$^5$ cells (i.v. injection) were used to engraft mice for experiments. Upon sacrifice due to AML, control spleen preparations were frozen for later use in the experiments designed to determine AML/BM toxicity. 8-9X10$^5$ cells were i.v. injected into non-conditioned mice for these experiments. The AE46T cell line was originally established by sequential retroviral transduction of UCB CD34+ cells with cDNAs encoding AML/ETO and hTERT [29, 30] and was maintained in IMDM/20%FBS supplemented with 10ng/mL SCF, TPO, FLT3-L, IL-3, and IL-6. 1X10$^6$ cultured cells were injected i.v. to induce AML.

Patient samples were obtained from patients at Cincinnati Children's Hospital Medical Center following informed written consent of parents/guardians and assent of patients over 11 years old. Residual diagnostic specimens were used according to a study protocol (#2008–0021) approved by the Cincinnati Children's Hospital Institutional Review Board (Office for Research Compliance and Regulatory Affairs). Additionally, we used a pre-existing PDX model (frozen spleen from secondary engrafted mice) which was previously generated from

cells from a deidentified sample (DFAM-64519-V2, PRoXe.org [31]). Initial primary specimens were incubated with OKT3 antibody to eliminate the potential for xenogeneic GVHD [26]. Following successful engraftment, BM and spleen preparations from primary mice were viably frozen for future experiments. 1–3 X10$^6$ thawed cells were injected i.v. for the PDX experiments described in this study. All cell transplants in this study were done by i.v. injection. A table with patient sample and PDX model information is included as supplementary material (S1 Table).

## Chemotherapy

A single i.p. dose of 30mg/kg busulfan was used as a preconditioning regimen in some experiments as described previously [26, 32]. For AML therapy, mice received 1.2mg/kg daunorubicin (D) and 50mg/kg cytarabine (A, Ara-C) by i.v. injection for 3 consecutive days beginning 2–3 weeks after busulfan conditioning and/or cell engraftment (low dose DA therapy). DA was repeated for some mice. Alternatively, a higher dose of 3.0mg/kg daunorubicin and 75mg/kg Ara-C was used (high dose DA therapy). For some experiments, doxorubicin was substituted for daunorubicin. For B-ALL 4-drug induction therapy, we used a 4-week schedule of vincristine (V, 0.5mg/kg, i.p., each Monday), dexamethasone (X, 15mg/kg, i.p., each day Monday-Friday), pegaspargase (P, 1200kU/kg, i.p., 1$^{st}$ and 3$^{rd}$ Monday), and daunorubicin (D, 2.5mg/kg, i.v., each Monday). This treatment is abbreviated "VXPD". When optimizing chemotherapy doses, mice were monitored for at least 6 weeks after exposure in an attempt to detect longer-term toxicities.

## PB and BM analysis

Tail bleeds were analyzed on a Hemavet9500 (Drew Scientific). Engraftment was determined from flow cytometry of PB and BM preparations using a FACSCantoII instrument (BD) with analysis by FlowJo software. Our standard flow panel consists of antibodies to block mouse and human Fc IgG receptors (Miltenyi Biotech) as well as mCD45-APC/Cy7(BD), CD45-FITC (BD), CD3-PE/Cy7 (BD), CD19-VioBlue (Miltenyi Biotech), CD13-PE (BD), CD33-PE (BD), CD34-APC (BD), and CD56-v510 (BD). Leukemia percentage was determined by calculating the number of cells with positive staining for CD33 and CD45 (AML) or CD19 and CD45 (ALL) as a fraction of viable cells.

## Statistics

Statistics were calculated with Prism 7 software. The Mann-Whitney U test was used to compare 2 groups. 2-way ANOVA was used to compare groups with repeated measurements. Log-rank analysis was used to compare survival curves. Linear regression analysis was performed to compare trendlines.

# Results

## RAG mice tolerate higher doses of AML chemotherapy

We began our comparison of SCID and RAG-based mice by searching for the maximum tolerated dose for combined daunorubicin and Ara-C intravenous infusions over three consecutive days. Our initial chemotherapy model utilized 5 days of Ara-C, however, we found that Ara-C alone produced very little leukemia cell specific killing benefit while adding measurable normal BM toxicity [6]. Therefore, we eliminated the final 2 days of Ara-C exposure. We also switched the anthracycline, replacing doxorubicin with daunorubicin in order to better mimic pediatric AML therapy protocols. NSGS (SCID) mice experienced lethal toxicities at all doses

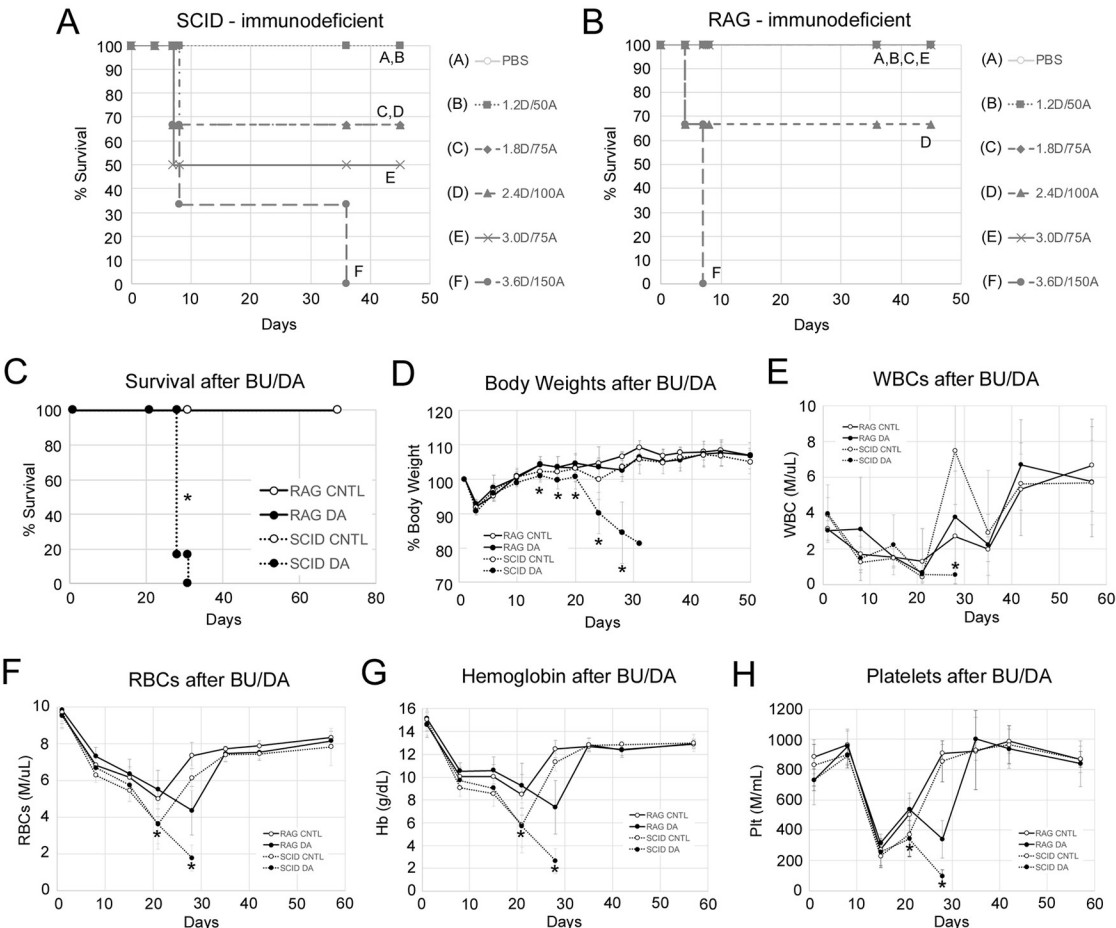

**Fig 1. RAG knockout mice tolerate higher AML therapy than SCID-based mice.** A) Naïve male NSGS mice (n = 3–4, 25.6 +/- 1.1g average body weight) with SCID immune deficiency or B) NRGS male mice (n = 3–4, 27.8 +/- 2.3g) with RAG knockout immune deficiency were challenged with various doses (doses in mg/kg) of combined daunorubicin (D) and Ara-C (A) injection for 3 consecutive days. Survival was monitored to determine maximum tolerable doses. Mice were sacrificed when they reached humane endpoints as described in the methods. (C-H) Mice (n = 4–5 per group) were conditioned with busulfan 3 weeks before exposure to 1.2mg/kg daunorubicin and 50mg/kg Ara-C (BU/DA). Survival (C), relative body weight (D), PB WBCs (E), RBCs (F), hemoglobin (G), and platelets (H) were monitored for responses to chemotherapy. For C-H, 8–12 week old female mice were used with starting weights of 25.4 +/- 3.0g (NRGS) and 24.4 +/- 1.9g (NSGS). Asterisks indicate p<0.05. For C-H, asterisks indicate significant differences between the SCID DA and RAG DA groups. CNTL = PBS controls, DA = combined daunorubicin/Ara-C, WBCs = white blood cells, RBCs = red blood cells.

higher than 1.2 mg/kg daunorubicin and 50mg/kg Ara-C (Fig 1A) in line with our previous findings. NRGS (RAG) mice survived a 50% higher dose but succumbed to a double dose of 2.4mg/kg daunorubicin and 100mg/kg Ara-C (Fig 1B). NRGS mice were resistant to increasing daunorubicin to 3.0mg/kg if Ara-C remained at 75mg/kg, implying that Ara-C may contribute more substantially to off-target toxicities. These results establish that RAG mice tolerate substantially higher chemotherapy doses.

Next, we exposed NSGS and NRGS mice to sub-lethal busulfan doses 3 weeks prior to chemotherapy to mimic an approach requiring preconditioning for successful engraftment of leukemia. Preconditioning is required for reliable engraftment of many samples and can significantly speed up disease latency for most. SCID and RAG strains received 1.2mg/kg daunorubicin and 50mg/kg Ara-C for 3 consecutive days. Previously we showed that NSGS mice cannot tolerate a similar 5+3 doxorubicin/Ara-C protocol after either irradiation or busulfan

conditioning [6]. Consistent with those findings, the NSGS busulfan+DA group experienced lethal toxicities several days after chemotherapy while similarly-treated NRGS mice survived for the duration of the 5-week post chemotherapy observation period (Fig 1C). The NSGS busulfan+DA group experienced more profound weight loss and failure to recover WBC, RBC, and PLT counts while these parameters returned to baseline levels in NRGS (Fig 1D–1H).

### Optimization of DA therapy in NRGS PDX mice

Next, we sought to test the efficacy of combined daunorubicin and AraC in leukemic NRGS mice, with and without prior busulfan conditioning. For this, we used a paired set of de novo and relapse PDX samples from the same patient. Busulfan conditioning was used to aid engraftment of the de novo sample but not the relapse sample. The lower dose of 1.2mg/kg daunorubicin and 50mg/kg AraC (Low Dose, LD) was used because that was the dose success-fully tested with busulfan conditioning in Fig 1. Marrow aspirates taken after therapy showed significantly decreased AML levels in the mice harboring the de novo sample, but not in those engrafted with the relapse sample (Fig 2A). However, this effect did not translate into increased survival in the de novo group (Fig 2B). Similarly, DA treatment did not affect survival of the mice with the relapse sample either (Fig 2C).

One possibility for the disconnect between initial treatment response and survival time is that the treatment damages both normal and leukemic cells which then compete to repopulate the bone marrow. If the AML is not sufficiently repressed, then the remaining cells may expand rapidly after therapy and effectively eliminate the gap in AML burden between the treated and control cohorts. Another possibility is that daunorubicin is not as effective as doxorubicin in PDX models. We tested both anthracyclines in two separate approaches to address this lack of efficacy.

First, we tested whether multiple cycles of LD chemotherapy would be tolerated in NRGS mice and improve survival. We engrafted NRGS mice with MA9.3Ras cells and initiated che-motherapy at day 10. After a 1-week break, some mice received a second round of chemother-apy. Others went on to receive a third round according to the same schedule. This schedule of repeated cycles more closely resembles typical patient therapy, which calls for additional ther-apy in MRD+ or high-risk cases. NRGS mice tolerated additional chemotherapy cycles and

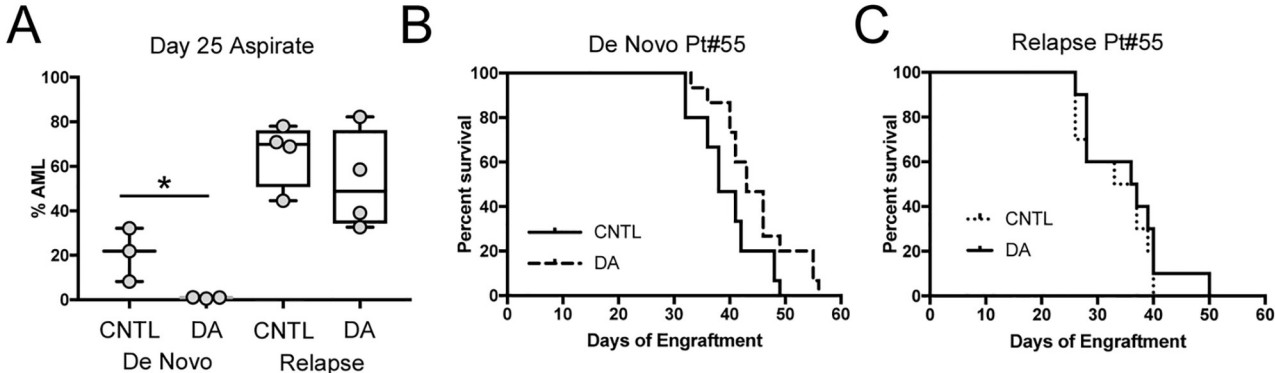

**Fig 2. Lack of efficacy with DA in AML-engrafted NRGS.** A) NRGS mice were engrafted with PDX samples generated from a paired de novo/relapse AML case. Busulfan was used to pre-condition mice for de novo engraftment, but not for mice receiving the relapse sample. Mice were treated with 1.2mg/mL daunorubicin and 50mg/kg Ara-C at 3 weeks and BM aspirates were analyzed at day 25. Survival of the mice engrafted with the B) de novo and C) relapse PDX samples was monitored. Asterisk indicates p<0.05 by Mann-Whitney U test. CNTL = PBS controls, DA = combined daunorubicin/Ara-C. Mice were randomly assigned to treatment or control groups.

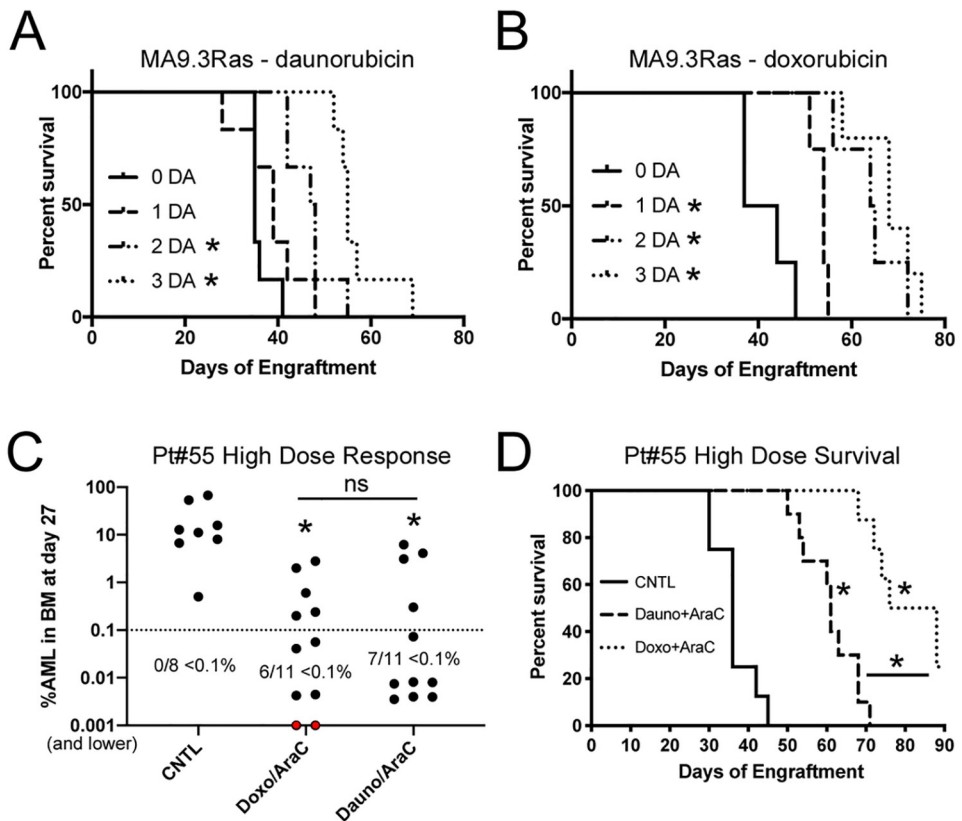

**Fig 3. Comparison of doxorubicin to daunorubicin in AML PDX models.** A) Survival of NRGS mice engrafted with the MA9.3Ras cell line and treated with 0, 1, 2, or 3 cycles of 1.2mg/kg daunorubicin and 50mg/kg Ara-C (0 DA, 1 DA, 2 DA, 3 DA) or B) 1.5mg/kg doxorubicin and 50mg/kg Ara-C. C) A de novo PDX was engrafted into NRGS mice and treatment began 3 weeks after busulfan conditioning and engraftment using HD DA using either 3.0mg/kg doxorubicin or daunorubicin. AML burden was determined from BM aspirates at day 27. The red points indicate mice with undetectable disease and are plotted as 0.001 in order to include them in the log based plot. D) The mice in C were followed for survival. Log rank tests were used for A,B,D. Mann-Whitney U tests were used to determine significance for C. Asterisks indicate p<0.05. CNTL = PBS controls. Mice in A-D were randomly assigned to treatment or control groups.

survived longer with each successive round of therapy. Consistent with our previous results, a single cycle of 1.2mg/kg daunorubicin and 50mg/kg Ara-C did not show efficacy (Fig 3A). However, when doxorubicin was substituted for daunorubicin, a statistically significant extension of latency was observed with a single cycle which was also further improved by additional cycles (Fig 3B).

Secondly, since we found that RAG mice could tolerate higher chemotherapy doses, we treated NRGS mice engrafted with a chemo naïve PDX sample with 3.0mg/kg daunorubicin or doxorubicin and 75mg/kg AraC (High Dose, HD). NRGS mice tolerated this higher chemotherapy dose 21 days after busulfan conditioning. Mice treated with either HD daunorubicin or doxorubicin (at the same dose) exhibited similar AML burden after therapy (Fig 3C). Approximately half of the mice in each group had AML at less than 0.1% by flow, a clinical cut-off for MRD status. However, most mice did relapse, although survival time was significantly extended (Fig 3D). Notably, doxorubicin resulted in a greater extension of lifespan compared to daunorubicin. In fact, 2 of the 11 HD doxorubicin treated mice had no detectable disease at the end of the experiment.

### Use of HD daunorubicin/AraC in de novo and relapse PDXs

We tested the optimized HD chemotherapy treatment protocol in our paired de novo / relapse PDX set. Mice engrafted with the de novo sample responded to therapy with a significantly longer latency while the relapse-engrafted mice showed no response to therapy (Fig 4A). In addition, the HD chemotherapy but not the LD protocol extended the lifespan of busulfan conditioned NRGS mice engrafted with a second chemotherapy-naïve sample (Fig 4B).

We also tested the HD chemotherapy response of mice engrafted with the chemotherapy-naïve AE46T cell line which was derived from UCB CD34+ cells with retroviral directed expression of *RUNX1/RUNX1T1* (*AML1/ETO*) and *TERT* [29]. HD chemotherapy was initiated at day 46, after engraftment was confirmed in the busulfan preconditioned recipients, resulting in delayed progression of leukemia (Fig 4C). We attempted to further delay leukemia by re-treatment at day 110, however the treated mice experienced significant toxicities and the experiment was ended. We repeated this approach with conditioned mice engrafted with a refractory adult MDS/AML sample with the first round of therapy at day 25 followed more closely by a second round of HD DA 2 weeks later. This timing resulted in mice with low tumor burden after therapy and increased lifespan (Fig 4D and 4E). Importantly, treated mice had similar AML in the BM at sacrifice as controls (CNTL, 64.0 +/- 5.9% vs DA, 72.0 +/- 12.1%), suggesting these mice succumbed to leukemia rather than treatment-related toxicities. Together, these results suggest that treatment toxicities increase in severity as tumor burden increases in PDX models.

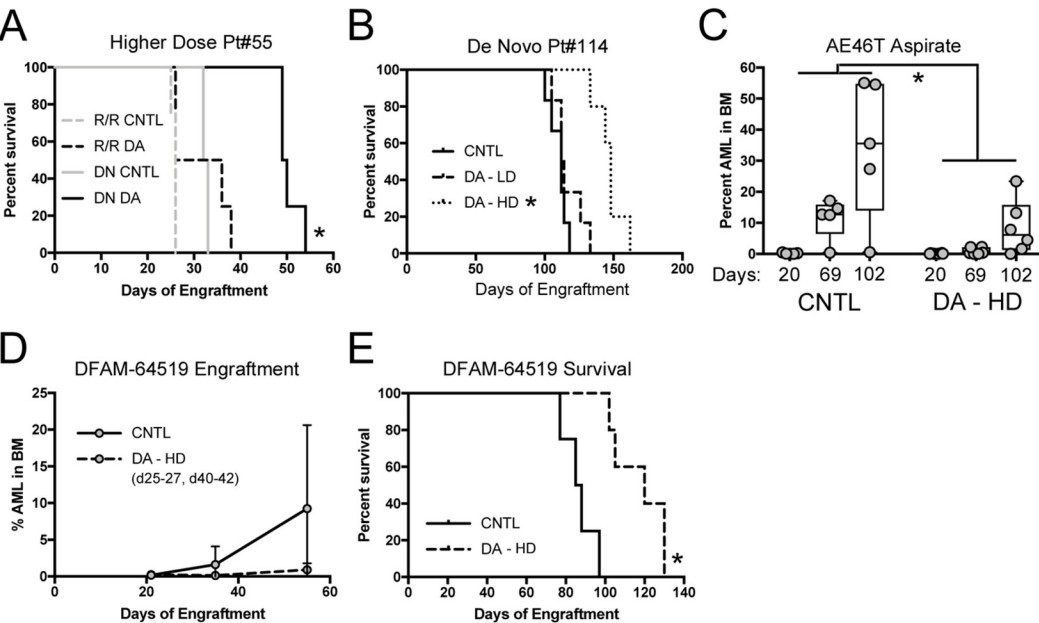

**Fig 4. HD chemotherapy for de novo and relapse AML PDXs.** A) Survival of NRGS mice engrafted with a matched de novo (DN)–relapsed/refractory (R/R) patient sample were treated with the higher dose of 3.0mg/kg daunorubicin and 75 mg/kg Ara-C. B) Survival of NRGS mice engrafted with a PDX sample from a second de novo case and treated with 1.2mg/kg daunorubicin and 50mg/kg AraC (DA-LD) or a higher dose (DA-HD) as in A. C) NRGS mice engrafted with the AE46T cell line were monitored for AML response to HD DA treatment. D) A relapse adult sample was subjected to two rounds of HD DA chemotherapy. BM AML burden and E) survival are shown. Asterisks indicate p<0.05 by log rank test (panel A, B, E), or 2-way ANOVA (panel C, D). Comparisons are treated versus controls. CNTL = PBS controls. Mice were randomly assigned to treatment or control groups.

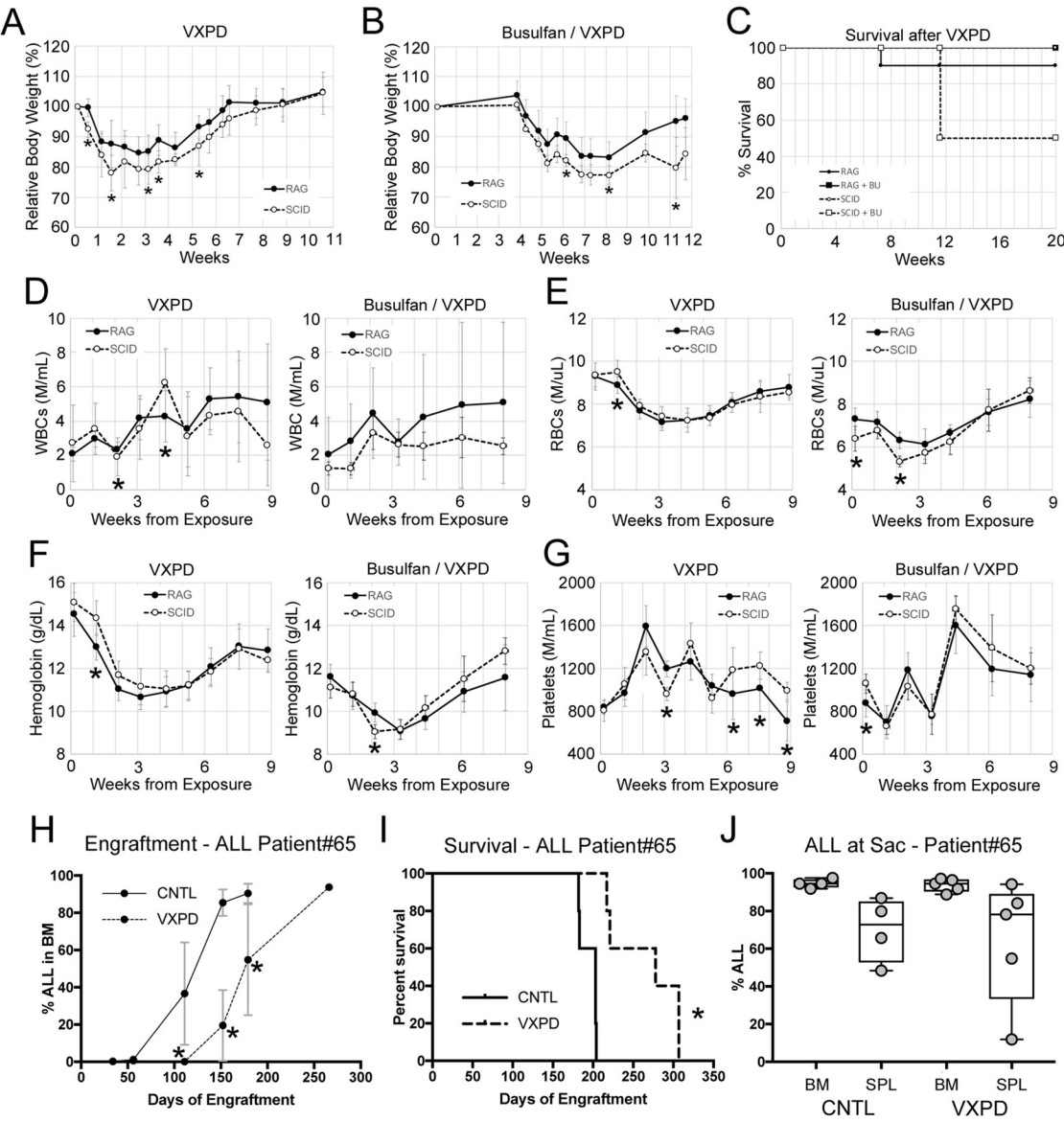

**Fig 5. Modeling 4-drug induction for high risk B-ALL.** A) Weights of NRG (RAG, n = 10) and NSG (SCID, n = 6) mice were treated with a 4-week course of VXPD. B) Weights of mice conditioned with busulfan 3 weeks prior to VXPD (n = 6 per group). C) Survival of mice in A and B. Mice were sacrificed when they reached humane endpoints as described in the methods. D) WBCs, E) RBCs F) hemoglobin, and G) platelets were monitored before, during, and after VXPD. H) PDX samples from a pediatric B-ALL sample were engrafted into NRG mice after busulfan conditioning. BM aspirates were analyzed by flow cytometry to monitor engraftment. I) Survival and J) B-ALL levels at time of sacrifice were determined for these mice. Asterisks indicate p<0.05 by Mann-Whitney U test (panels A-B, D-H, J) or log rank (panel I). CNTL = PBS controls.

## RAG mice better tolerate an ALL 4-drug induction protocol

To explore the suitability of RAG-based mice for B-ALL modeling, we examined the durability of NSG and NRG mice to a 4-drug induction protocol for high risk B-lymphoid leukemia. To test for tolerance, we initially exposed non-conditioned, non-leukemic RAG and SCID based mice to vincristine, dexamethasone, pegaspargase, and daunorubicin. SCID-based NSG mice experienced a more dramatic weight loss relative to RAG-based NRG mice, but both strains recovered from the 4-week treatment (Fig 5A). However, when busulfan conditioning was

included 3 weeks prior to chemotherapy, half of the SCID-based mice experienced lethal toxicities several weeks post exposure (Fig 5B and 5C). There were no obvious or consistent statistically significant alterations in hematopoietic parameters as measured by CBC analysis, indicating that this effect was unlikely to be related to excessive BM damage or failure and points to non-hematopoietic toxicity (Fig 5D–5G).

To test the efficacy of 4-drug ALL induction, a chemotherapy-naïve B-ALL was engrafted into busulfan conditioned NRG mice. The 4-week treatment started once B-ALL was detectable in the PB. Serial BM aspirations revealed a dramatic decrease in ALL burden in treated mice relative to controls (Fig 5H) which resulted in a significant latency shift (Fig 5I). Importantly, treated mice showed the same level of ALL as control mice at the time of sacrifice, indicating that the mice did not experience treatment-related toxicities (Fig 5J). These data demonstrate the utility of NRG mice in the modeling of high risk 4-drug ALL induction therapy.

## RAG mice offer a better therapeutic window for genotoxic agents

To this point, we have established that SCID mice have lower tolerance for chemotherapy regimens. We assume that this is at least partly due to the $Prkdc^{SCID}$ mutation being consequential in all cells. DNA repair should be compromised in SCID mice and therefore we would expect higher rates of apoptosis in response to DNA damaging agents. This problem should be avoided in RAG mice, because RAG knockout should specifically affect lymphocyte development and play no direct role in DNA damage response. The MA9.3Ras cell line causes fatal AML in both NSGS and NRGS with very similar kinetics (Fig 6A), making this model suitable for testing this hypothesis, and for quantifing any SCID/RAG differences. To examine initial response to chemotherapy, we subjected non-conditioned mice engrafted with MA9.3Ras cells to 3 consecutive days of DA exposure and sacrificed them 3 days later. For this experiment we used the SCID MTD of 1.2mg.kg daunorubicin and 50mg/kg AraC (LD). In NSGS DA treated mice, BM cellularity was reduced to 34.7% of controls (25.4 $X10^6$ vs 8.8 $X10^6$ WBCs/femur) while a somewhat smaller decrease was observed in NRGS (31.4 $X10^6$ to 13.7 $X10^6$ cells, 43.7%) (Fig 6B). NRGS DA mice had significantly more surviving BM cells than NSGS DA mice (p = 0.0073), however, increased NRGS BM cellularity was also noted in control mice. This finding might be at least partially explained by the overall larger size of age-matched male NRGS compared to NSGS mice (32.0+/-2.3g vs 28.5+/-2.0g, p = 8E-06, Fig 6C). Linear regression analysis of age/weight data confirmed that the NRGS mice used in these experiments were larger than their SCID-based counterparts. Separate analysis of NRG and NSG showed similar significant differences in both age-matched males and females (not shown). Absolute AML cells per femur was decreased in both strains in response to chemotherapy. However, this decrease was more dramatic in NRGS mice, where DA-treated mice contained on average only 0.34%+/-0.26% of control levels while NSGS DA-treated mice retained 0.96%+/-0.73% (p = 0.027, Fig 6D). By taking the percent decrease of normal mouse BM and human AML cell numbers in response to DA together, we calculated the relative AML specific toxicity in both strains. For NRGS mice, DA treatment resulted in a 49.5-fold decline in absolute AML number compared to normal mouse BM cells while NSGS mice experienced only a 13.1-fold difference, suggesting a larger therapeutic window (a 3.8 fold difference between strains) for cytotoxic chemotherapy in NRGS mice (Fig 6E).

## Discussion

We show that RAG-based mice tolerate a busulfan preconditioning regimen in combination with leukemia chemotherapy, expanding the number of samples that can be studied in

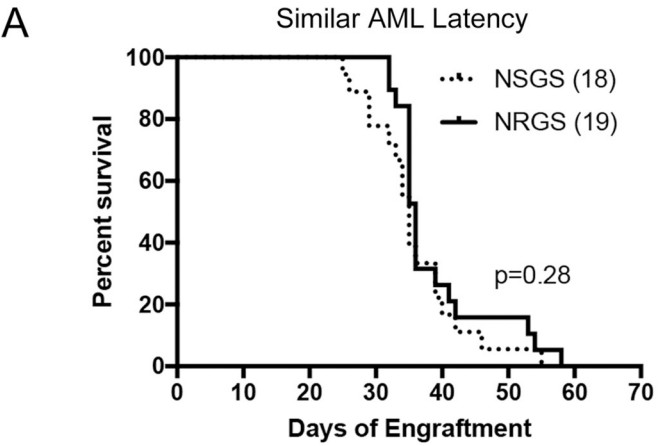

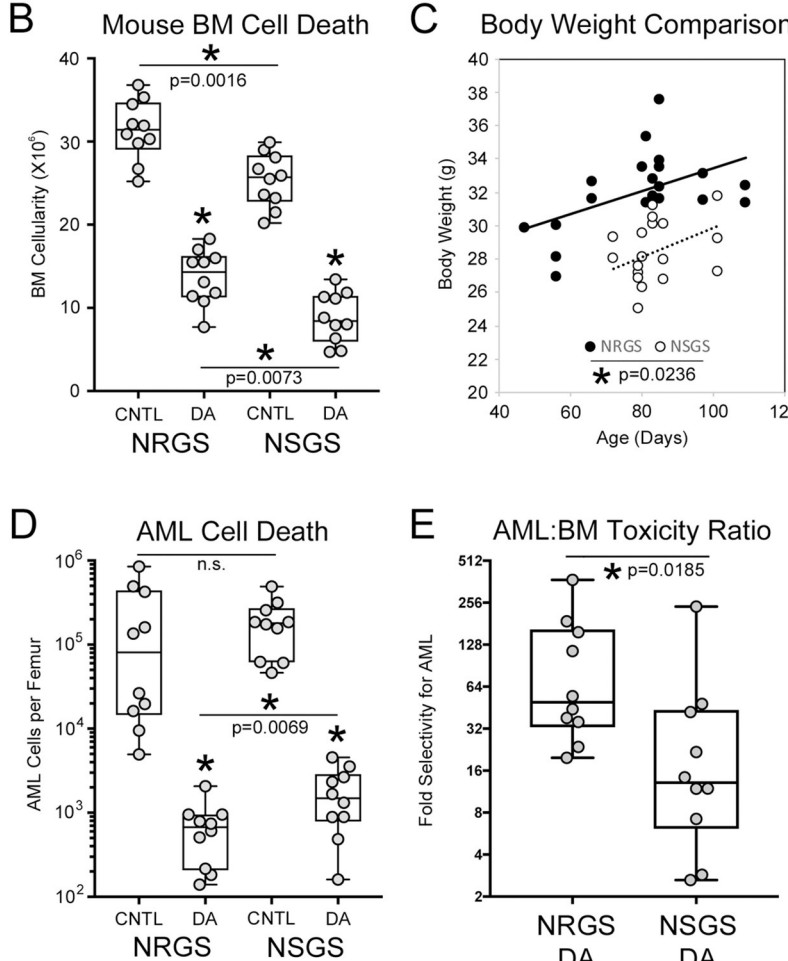

**Fig 6. Determination of therapeutic window in NSGS and NRGS mice.** A) Latency of MA9.3RAS leukemia induction in NSGS and NRGS mice. B) Day 14 BM cellularity of mice engrafted with MA9.3RAS cells. Mice were treated with 3 days of PBS (CNTL) or DA (1.2mg/kg daunorubicin and 50mg/kg Ara-C) beginning on day 9. Mice were randomly assigned to treatment or control groups. C) Body Weight plotted against age of mice used in panel B. D) Absolute MA9.3RAS cell number in mice presented in panel B. E) The toxicity ratio was calculated for each DA treated mouse. Log rank test was used for panel A. Asterisks indicate p<0.05 by Mann-Whitney U test (panel B, D, and E) or linear regression analysis (panel C).

vivo. RAG-based mice also tolerate multiple cycles of therapy, thereby allowing for more aggressive, realistic modeling. Furthermore, standard AML therapy in RAG mice was 3.8-fold more specific for AML cells, relative to SCID mice, demonstrating an improved therapeutic window for genotoxic agents. We conclude that RAG-based mice represent the new standard for preclinical evaluation of therapeutic strategies involving genotoxic agents.

We did not cure any mice using either our daunorubicin HD or repeated cycle protocols, even in a PDX model of a de novo patient that achieved a MRD(-) remission clinically. This suggests that while we have improved modeling of standard therapy, the models are still not optimized. One difficulty is likely the relative lack of benefit from Ara-C in these models. Ideally, Ara-C would be delivered as a continuous slow infusion by implanted osmotic pumps over the course of a week rather than as several bolus injections. Alternatively, liposomal formulations could also improve efficacy [33]. Supportive care to combat treatment toxicities is also largely absent in PDX models and this may limit the successful implementation of any protocol, particularly if disease burden is high. However, further optimization may not be the preferred approach, since a perfected model would likely make it more difficult to realize a PDX benefit from additional novel therapies.

The ability to tolerate higher chemotherapy doses also suggests that additional cytotoxic agents could be added to the DA backbone. For example, pediatric AML patients are commonly treated with etoposide in addition to DA (so-called ADE induction therapy). The RAG background should allow for expanded chemotherapy modeling in mice. Additionally, targeting anti-apoptotic Bcl-2 familiy proteins along with standard therapy is of great interest, however these inhibitors have been shown to sensitize to anthracyclines [34–36] posing a major challenge for in vivo experiments with SCID mice. The SCID defect would also cause sensitivities to other therapies that induce double strand DNA breaks. As a result, experiments that couple proton therapy to chemotherapy or other sensitizers [37] could be more easily done with RAG-based mice.

The finding of a 3.8-fold improvement of the therapeutic window with RAG relative to SCID mice is similar to the 2–3 fold difference in sensitivity reported between SCID and BALB/c fibroblasts exposed to bleomycin or gamma irradiation, both of which induce double strand DNA breaks [16]. Anthracyclines such as daunorubicin and doxorubicin inhibit the ability of Topoisomerase II to reseal double strand DNA breaks. Given that the SCID mutation renders cells defective in double strand break repair, this likely provides the rationale for increased sensitivity of SCID mice to both the DA and VXPD induction protocols and a worse therapeutic window.

Interestingly, we have found that SCID and RAG mice have similar sensitivities to busulfan conditioning but react differently upon additional genotoxic stress. Busulfan works by induction of intra-strand crosslinks and mono-alkylation of DNA [38], so one might predict repair to be independent of the $Pkrdc^{scid}$ mutation. However, prior exposure to busulfan further increased the sensitivity of SCID mice to ds-DNA break inducing agents as evidenced by the failure of conditioned NSGS mice to tolerate DA therapy at doses tolerated by naïve NSGS mice. This has clear implications for studies that combine anthracyclines with other DNA damage inducing agents, even if the mechanisms of action are distinct. RAG-based immune-deficient mice should be used for chemotherapy modeling that requires conditioning of mice prior to engraftment.

Although the SCID-associated genotoxic sensitivities are especially severe when ds-DNA break-inducing agents are used, it should be appreciated that some agents that do not damage

DNA may increase sensitivity to anthracyclines. For example, the BCL inhibitor venetoclax enhances the effects of ionizing radiation [39]. Experimental MDM2 inhibitors are potent activators of p53 which could be expected to sensitize cells to anthracycline therapy [40, 41]. Furthermore, some compounds may produce unexpected toxicities in the same way, as has been described recently for abemaciclib [42]. These activities are likely to increase non-specific toxicities of standard chemotherapy in SCID-based mouse models, effectively limiting detection and measurement of a pre-clinical therapeutic window.

In the current study we substituted doxorubicin with daunorubicin in order to more closely follow accepted clinical protocols. Surprisingly, we found less efficacy than expected with daunorubicin and a marked improvement of response to doxorubicin over daunorubicin in our PDX models in head to head experiments. This could simply reflect a difference in human and mouse metabolism of the drugs. On the other hand, it could indicate a true difference in the efficacy of these anthracyclines. The optimal dose for individual anthracyclines is different for each drug and there is active study and debate about the relative efficacy between the members of the class. Non-hematopoietic toxicities are an important clinical consideration that must be balanced against the anti-leukemic effect. A retrospective study of childhood cancer survivors demonstrated that daunorubicin resulted in approximately half the risk of future cardiac failure relative to doxorubicin [43]. Doxorubicin has also been found to be associated with more complications due to infections than daunorubicin when given to ALL patients during delayed intensification [44]. Another study with retrospective analysis of a large group of patients found that in children over 3 years of age, doxorubicin was associated with significantly higher rates of induction related mortality, but fewer induction failures than were observed with daunorubicin [45].

Similarly, we have previously used L-asparaginase for PDX ALL induction therapy but switched to pegaspargase in order to update our models to more closely follow practices in pediatric oncology. A large multi-center trial of childhood de novo ALL found that results and toxicities from biweekly pegaspargase were very similar to those observed with weekly intramuscular injection of native L-asparaginase given after initial induction induced remission [46]. Similarly, a comparison in a relatively low number of adult high-risk ALL patients found no difference in clinical outcomes [47]. In a cohort of relapsed pediatric ALL patients, while pegaspargase demonstrated a prolonged half-life, there was an observed trend towards lower asparagine clearance in the CNS [48]. It remains to be seen whether this substitution has any effects in the PDX setting.

Recently, we have shown that humanized NSGS mice have improved hematopoietic function over humanized NSG mice [49]. Moving forward, it will be important to test immune function in NRGS mice as well, since these mice could be better hosts to build immune therapy models with, particularly if exposure to genotoxic agents is planned. The ability to busulfan condition prior to chemotherapy will be an important advantage for building better models of therapy. For example, conditioning is required for engraftment and humanization with UCB. UCB-transplanted mice might allow for the evaluation of therapies in the context of human immune cells [50].

## Supporting information

**S1 Table. Summary of PDX models.** Age, sex, and stage of disease of the source material used to build each PDX model is listed. The genomic alterations present in the PDX models as well as a description of the latencies in mouse strains with and without busulfan conditioning is also listed.
(XLSX)

## Acknowledgments

The authors thank the CCHMC Flow Cytometry Core for access to FACS machines and the CCHMC Comprehensive Mouse and Cancer Core for supplying some of the mice used in these experiments.

## Author Contributions

**Conceptualization:** Mark Wunderlich, Benjamin Mizukawa, James C. Mulloy.

**Data curation:** Mark Wunderlich, Nicole Manning, Christina Sexton, Anthony Sabulski, Luke Byerly, Eric O'Brien.

**Formal analysis:** Mark Wunderlich.

**Funding acquisition:** Mark Wunderlich, John P. Perentesis, James C. Mulloy.

**Investigation:** Mark Wunderlich, Nicole Manning, Christina Sexton, Anthony Sabulski, Benjamin Mizukawa.

**Methodology:** Mark Wunderlich, Benjamin Mizukawa, James C. Mulloy.

**Resources:** Luke Byerly, Eric O'Brien, John P. Perentesis, James C. Mulloy.

**Supervision:** Mark Wunderlich, John P. Perentesis, James C. Mulloy.

**Writing – original draft:** Mark Wunderlich.

**Writing – review & editing:** Mark Wunderlich, Benjamin Mizukawa, James C. Mulloy.

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
