## [Decision Letter · Decision Letter 0]

10 Oct 2019

PONE-D-19-20568

Improved chemotherapy modeling with RAG-based immune deficient mice

PLOS ONE

Dear Mr. WUNDERLICH,

Thank you for submitting your manuscript to PLOS ONE. Apologies for delay - it was difficult to find timely reviewers with appropriate expertise. After careful consideration, we feel that it has merit but does not fully meet PLOS ONE’s publication criteria as it currently stands. Therefore, we invite you to submit a revised version of the manuscript that addresses the points raised during the review process.

We would appreciate receiving your revised manuscript by Nov 24 2019 11:59PM. To enhance the reproducibility of your results, we recommend that if applicable you deposit your laboratory protocols in protocols.io, where a protocol can be assigned its own identifier (DOI) such that it can be cited independently in the future. For instructions see: http://journals.plos.org/plosone/s/submission-guidelines#loc-laboratory-protocols

We look forward to receiving your revised manuscript.

Kind regards,

Daniel Thomas, MD

Academic Editor

PLOS ONE

Journal Requirements:

1. Thank you for inclouding your funding statement; "This work was funded in part by an NIH/NCI R50 award (#CA21140, MW). The funders had no role in study design, data collection and analysis, decision to publish, or preparation of the manuscript."

Additional Editor Comments (if provided):

Reviewers' comments:

Reviewer's Responses to Questions

**Comments to the Author**

1. Is the manuscript technically sound, and do the data support the conclusions?

Reviewer #1: Yes

Reviewer #2: Yes

2. Has the statistical analysis been performed appropriately and rigorously? 

Reviewer #1: Yes

Reviewer #2: Yes

3. Have the authors made all data underlying the findings in their manuscript fully available?

Reviewer #1: Yes

Reviewer #2: Yes

4. Is the manuscript presented in an intelligible fashion and written in standard English?

Reviewer #1: Yes

Reviewer #2: Yes

5. Review Comments to the Author

Reviewer #1: An extremely valuable piece of work at a time when many academic institutions and industry are expanding their patient-derived xenograft models. The superior robustness of NRG and NRGS mice to tolerate high doses, repeated doses and busulfan pre-conditioning has been suspected by many in the field but never definitively compared with the same patient samples/ cell lines and conditions. The improved efficacy of doxorubicin compared to daunorubicin for AML is also intriguing and correlates with increased toxicity observed in humans. The calculation of AML vs normal murine cell ratio comparing NSG and RAG mice is an informative measurement; if only similar studies were performed by other groups.

English and figure presentation is excellent.

Minor points

1] It is not clear as you read how old the mice are after weaning, in general, or specifically, at the time of engraftment and conditioning. Sex and weight of mice in Figure 1 are also not specified.

2] A small table of patient details for the de novo and relapse pairs would be useful for other groups to compare their own rates of engraftment.

3] I assume all samples were tail vein engrafted but BM sampling was with live aspiration - this was not completely clear in methods.

4] The lack of busulfan conditioning in the relapsed sample Fig 2A makes it difficult to assess the lack of reponse to DA. Any other data to show what happens to relapsed sample with bu conditioning? Any data to show complete lack of engraftment of de novo without any conditioning?

5] Fig 2 CNTL not obvious to every reader - please define in figure legend. Busulfan not stated as given in the Figure legend - this is important variable.

6] A similar indice comparing engrafted AML response to chemo to engrafted normal human cord blood derived hematopoiesis response to chemo would be interesting to extrapolate and compare with clinical trials.

Reviewer #2: The manuscript follows, broadly, a similar approach to earlier publications by these authors, and establishes the utility of RAG-based immunodeficient mice as a model fro AML PDX. The work presented in this manuscript is sound, and well performed, and will be a useful resource for others working in the field. I would judge that there a number of groups that have perhaps already reached similar conclusions as those resented in this manuscript. Nevertheless, this is a paper that warrants publication, as it presents a solid characterisation of the model and response to standard chemotherapeutic regimens. I do not have any major changes to suggest. I would argue that there needs to be some attention paid to the presentation of the figures. In many instances, the acronyms used in the figures are not described in the figure legends. For example in Figure 1 BU/DA is not explained (I accept that one can guess easily enough) or there is somewhat eclectic title to a figure panel such as 6 C. However, beyond these very minor suggestions, i judge the manuscript will find a readership in those interested in AML PDX models largely as it is written.

6. PLOS authors have the option to publish the peer review history of their article (what does this mean?). If published, this will include your full peer review and any attached files.

Reviewer #1: Yes: Daniel Thomas

Reviewer #2: No

---

## [Author Response · Author response to Decision Letter 0]

25 Oct 2019

We thank the reviewers for their time and careful consideration of our manuscript. Please find the responses to individual points below. Changes to the manuscript are indicated using the track change feature in Word.

Reviewer #1: An extremely valuable piece of work at a time when many academic institutions and industry are expanding their patient-derived xenograft models. The superior robustness of NRG and NRGS mice to tolerate high doses, repeated doses and busulfan pre-conditioning has been suspected by many in the field but never definitively compared with the same patient samples/ cell lines and conditions. The improved efficacy of doxorubicin compared to daunorubicin for AML is also intriguing and correlates with increased toxicity observed in humans. The calculation of AML vs normal murine cell ratio comparing NSG and RAG mice is an informative measurement; if only similar studies were performed by other groups.

Thank you for the positive review of our work. 

English and figure presentation is excellent.

Minor points

1] It is not clear as you read how old the mice are after weaning, in general, or specifically, at the time of engraftment and conditioning. Sex and weight of mice in Figure 1 are also not specified.

We have included additional information regarding age and sex in the methods and the specific information for figure 1 in the legend.

2] A small table of patient details for the de novo and relapse pairs would be useful for other groups to compare their own rates of engraftment.

We have included a small table (Table S1) with patient sample and PDX information.

3] I assume all samples were tail vein engrafted but BM sampling was with live aspiration - this was not completely clear in methods.

Yes, the reviewer is correct. We have made this clearer in the methods. See line 132 for aspirate information and 162 for i.v. injection.

4] The lack of busulfan conditioning in the relapsed sample Fig 2A makes it difficult to assess the lack of reponse to DA. Any other data to show what happens to relapsed sample with bu conditioning? Any data to show complete lack of engraftment of de novo without any conditioning?

Busulfan conditioning speeds up engraftment of almost every AML PDX. As it is a chemotherapy, busulfan conditioning is somewhat toxic, but the effects are transient, and mice fully recover after a couple of weeks, before chemotherapy (DA) exposure. This is why our protocol is to wait 3 weeks before chemotherapy treatment. By this time, prior busulfan exposure should not be a factor in the response to DA. With aggressive samples such as Pt#55 relapse, busulfan conditioning would likely cause the latency to fall before that three-week timepoint. We have seen this in previous work with cell lines that have similar latencies. Therefore, using busulfan for such an aggressive sample is not feasible. Even though the relapse sample was engrafted into non-conditioned mice, the aspirate data in figure 2A shows that engraftment is actually better than the presentation sample with busulfan pre-conditioning. We have seen previously that some samples do not engraft without conditioning. More frequently, conditioning allows for a more robust early engraftment and shorter latency of disease. This is an important tool which can decrease time and money spent on lengthy in vivo experiments involving samples with latencies that would otherwise be 100 days or longer. 

Our study only sought to determine conditions under which pre-conditioning could be combined with subsequent chemotherapy. This was important to us because pre-conditioning with irradiation or busulfan has well known benefits and is a common practice in PDX modeling. However, we did not design experiments to specifically demonstrate the benefits of conditioning as part of this study.

5] Fig 2 CNTL not obvious to every reader - please define in figure legend. Busulfan not stated as given in the Figure legend - this is important variable.

We have defined CNTL in the legend. We have also included a statement regarding busulfan conditioning in the legend to make this clear.

6] A similar indice comparing engrafted AML response to chemo to engrafted normal human cord blood derived hematopoiesis response to chemo would be interesting to extrapolate and compare with clinical trials.

We agree that this could be an interesting and potentially useful system to also measure the effects on normal human cells in RAG based xenografts. Perhaps it would be even more appealing to use the assay to test the activities of a novel experimental agent, in order to show pre-clinical specificity or potential for a therapeutic window.

However, this approach would require significant effort to establish and optimize as several technical considerations make this experiment very difficult to perform and interpret. First, significant engraftment of UCB CD34+ requires either very high cell numbers or busulfan conditioning. We could evaluate effects after reconstitution of the BM has stabilized and after busulfan effects on the BM are repaired, ~8-10 weeks later, but the engraftment levels and cell type subpopulations within a cohort of UCB CD34+ transplanted mice are quite variable at this time. This makes determination of total normal cells before and after therapy (averaging control / treated groups) very imprecise. This is made worse by the fact that human cell engraftment affects the remaining murine BM cells, including a progressive BM failure in NSGS and NRGS mice (Wunderlich, JCI Insight, 2016, https://insight.jci.org/articles/view/88181), thereby altering total cellularity. Also, human cells do not fully recapitulate normal hematopoiesis in mice, so a human BM graft would likely have altered sensitivities to chemo as well, especially given the lack of human RBC production by human cells. Another variable that would need to be considered and tested is the rate of repopulation of the BM by the UCB graft and residual mouse hematopoiesis. The timing of sacrifice and quantification after chemo exposure will greatly affect the results if human and mouse cells repopulate at different rates.

Alternatively, if we engrafted UCB CD34+ cells and treated early with chemotherapy, it is unclear how many CD34+ cells would be required for this approach. We transplanted 800-900k MA9.3Ras cells for the AML experiment and using similar numbers of CD34+ would require an entire UCB unit for each 1 or 2 mice, making this approach cost-prohibitive. Busulfan conditioning would also be required and we would be initiating chemo before our tested window of 3 weeks. In short, there are many logistical issues to be considered and weighed to reach reliable conclusions with such an approach.

Reviewer #2: The manuscript follows, broadly, a similar approach to earlier publications by these authors, and establishes the utility of RAG-based immunodeficient mice as a model fro AML PDX. The work presented in this manuscript is sound, and well performed, and will be a useful resource for others working in the field. I would judge that there a number of groups that have perhaps already reached similar conclusions as those resented in this manuscript. Nevertheless, this is a paper that warrants publication, as it presents a solid characterisation of the model and response to standard chemotherapeutic regimens. I do not have any major changes to suggest. I would argue that there needs to be some attention paid to the presentation of the figures. In many instances, the acronyms used in the figures are not described in the figure legends. For example in Figure 1 BU/DA is not explained (I accept that one can guess easily enough) or there is somewhat eclectic title to a figure panel such as 6 C. However, beyond these very minor suggestions, i judge the manuscript will find a readership in those interested in AML PDX models largely as it is written.

We thank the reviewer for the positive assessment of our work. We agree that while others may have reached similar conclusions, we hope that publishing our findings will help the vast majority of researchers who are not as experienced in the intricacies of PDX modeling. 

We have defined acronyms in the figure legends for most figures. Additionally, we have changed the figure 6C title to “Body Weight Comparison”.

Reviewer#3

Review for Wunderlich et al.

PONE-D-19-20568 

“Improved chemotherapy modeling with RAG-based immune deficient mice”

Overall recommendation: accept with revision

To the Editors:

This manuscript describes a new NRG-based mouse model of leukaemia that is able to tolerate a more realistic human AML chemotherapy regimen thus providing a superior in vivo model than the NSG-based models currently used. Additionally, the inability of current NSG/NSGS mouse models to tolerate pre-chemotherapy busulfan conditioning is also overcome increasing the number of PDX engraftments able to be modelled. The manuscript is well written, the data sound and the study would be of interest to the readers of the PLOS ONE. However, the manuscript would benefit from some minor alterations/clarifications which I have in the comments to authors below. 

To the Authors:

Overall comments to authors

This study describes a new NRG-based mouse model of leukaemia that is able to better tolerate treatment regimens currently used in the clinic that the more widely used NSG models. The manuscript is well written and the data and conclusions sound. However, some additional clarifications and amendments are required before I can support publication. Please address my questions and comments as per below.

Thank you for the positive feedback.

1. It’s not always mentioned with what the control mice have been treated (busulfan alone, busulfan + PBS?). Can this please be interrogated throughout?

We have now defined CNTL in each legend.

2. Fig 2A, 3C, 4D, 4F, 5H, 5J: How was %AML determined, what % positivity of which markers from the flow panel? Is %AML the % of human markers relative to mCD45 or absolute % of human markers?

Also, which marker/s was used to determine B-ALL cells in the blood (line 339)?

CD33 was used to label the AMLs and CD19 was used for ALLs. The models used here also all had human CD45 expression. We present the percentage of positive cells within the total viable cell fraction. A sentence has been added to methods to relay this point.

3. The methods say "Additionally, BM and PB samples were periodically taken from leukemic mice in order to ascertain the level of leukemic burden and to better predict the onset of illness.” Was there a threshold level of engraftment that mice had to reach prior to treatment initiation and if so what was it?

No, we did not establish a threshold for engraftment. Our PDX models were all tested and characterized prior to the study and showed reliable engraftment without failure. The above quoted sentence is in reference to measures taken that aid in our efforts to minimize discomfort and needless suffering of the mice. 

Line 288 states: Engraftment was confirmed in the busulfan preconditioned recipients before HD chemotherapy at day 46. However, in Fig 3AB it's not clear whether MA9.3Ras engraftment was confirmed prior to day 10 treatment initiation? Similarly, in Fig 3C,D it's also not clear whether PDX engraftment was confirmed prior to day 7 treatment initiation. Whether confirmation of engraftment prior to treatment initiation occurred (or not) needs to be explicitly stated throughout.

Figure legends 2,3,4, and 6 have been updated to indicate random assignment of mice to treatment groups. 

4. Can the number of days relating to the survival curve comparisons please be stated in the text with p-values included throughout? It's a little hard to estimate from the graphs eg: single cycle comparison in Fig 3B appears to be 50 vs 53 days which doesn't seem like it would be a “significant extension of latency”.

The asterisks indicate statistically significant differences only (p<0.05 by log rank). The main findings are that the latency shifts further to the right with each successive round of therapy and that overall, doxo has a more robust effect than dauno in these PDX models. We have inserted “statistically” into the text to avoid suggesting that a modest extension represents a significant (in terms of scale) biological effect. 

5. Fig 3C:There are 9 dots in the CNTL group, not 8 and 2x dots are on the dotted zero line so shouldn’t the numbers quoted be 2/9 <0.1%? The data may be better represented with a split y-axis.

We have re-analyzed and re-plotted these results with log scale in order to better show the data points with low disease burden. Upon this review, one of the low-level control points represented a mouse that was sacrificed after this BM check with an apparent mouse cell leukemia/lymphoma soon after the aspirate data was taken and was therefore censored from the experiment. The 2 red points in the doxo/arac column had undetectable disease. The legend has been edited to reflect this.

6. Line 271, 272: “AML burden …. was equally suppressed…” But the authors don't know if the AML level was suppressed in treated groups as there was no initial determination of %AML, only at day 27. Similarly, line 273 “less than 0.1% AML by flow” – the low level of assessable AML may be due to lack of initial engraftment rather than response to treatment.

These points may not be relevant once point 3 is addressed though.

We have rephrased this sentence to “Mice treated with either HD daunorubicin or doxorubicin (at the same dose) exhibited similar AML burden after therapy (Fig 3C).” This change avoids assuming that the levels were equivalent at the start of treatment (the mice were randomly assigned to groups). 

No control mice exhibit engraftment of less than 0.1% at this timepoint, therefore we attribute the lowered level of engraftment in the treated cohorts to the activity of the drugs.

7. For Fig 4A shouldn't the Log rank and not the M-W test (as indicated in the legend) have been applied?

Yes, thank you for your attention to detail. This has been corrected in the legend.

Also, does the asterisk in 4A denote significance of de novo DN treatment group compared with all other groups, or just the DN CNTL group? Please specify in the legend. Similarly for Fig 4B, what comparisons have been made and which are significant?

Only controls vs treated were analyzed. A clarification was added to the legend.

8. Fig 5: The error bars overlap for 12/17 of the data points denoted with asterisks, how can there be significant?

We are simply reporting the result of the Mann-Whitney calculation done in Prism software. We do not make any claims that the individual CBC points represent meaningful differences. In fact, in the text we offer the interpretation that “There were no obvious alterations in hematopoietic parameters measured by CBC analysis, indicating that this effect was unlikely to be related to BM failure (Fig 5D-G).” We have edited this sentence to reinforce that idea. We say this because for each of the parameters (WBC, RBCs, Hg, Plt) there are only sporadic timepoints with p <0.05 and many of these are contradictory (such as PLT sometimes higher, sometimes lower in SCID relative to RAG, Fig 5G). The weights are likely to be more meaningful. For those measurements, the SCID line tracks continuously below the RAG line with recurrent timepoints of p<0.05. Also, extreme weight loss by some SCID mice was a key factor leading to the necessary sacrifice of mice in Fig 5B.

9. Lines 354-357: If MA9.3Ras engrafted mice were treated and all animals sacrificed 3 days later, how was the survival curve plotted in Fig 6A obtained? Please clarify or reword these sentences.

Figure 6A are results from a separate, initial experiment comparing MA9.3Ras cells in NSGS/NRGS. We agree that these sentences were a bit out of order. We have made edits to this paragraph to attempt to improve the flow of information.

Minor Comments

10. Spelling error, line 73 “agenet”

Please also define that UBC is (I presume) umbilical cord blood, line 286

Line 359: ‘M’ is not standard nomenclature for cell numbers, please use x10^6. Please alter on Fig 6B as well

Thank you, these corrections have been made.

11. Can more details for the cell lines used in the study please be given for readers who may not be familiar: MA9.3RAS, is this an AML cell line and from where was it obtained? The AE46T cell line is not mentioned in the methods section either

Our lab has generated both of these cell lines through retroviral transduction of UCB-CD34+ cell cultures. We have added references along with some information in the methods section and in the newly added Table S1.

12. Fig 1 legend: the description of D-H are a bit out of order. C=survival, not E. Please check the parameters on the graph and assigned figure letters in the legend

Thank you. This has been corrected.

13. Line 368: what is the p-value of the DA-treated vs control mice absolute cell number comparison?

The Mann-Whitney p value is 0.0073. We have added this to the text, line 369.

14. Fig 6B: what does the “m” in mBM represent?

We have changed Fig 6B to read as “mouse BM Cell Death” to make this clear.

---

## [Editor Report · Decision Letter 1]

7 Nov 2019

Improved chemotherapy modeling with RAG-based immune deficient mice

PONE-D-19-20568R1

Dear Dr. WUNDERLICH,

We are pleased to inform you that your manuscript has been judged scientifically suitable for publication and will be formally accepted for publication once it complies with all outstanding technical requirements.

With kind regards,

Daniel Thomas, MD

Academic Editor

PLOS ONE

Additional Editor Comments (optional):

All points have been adequately addressed.
---

## [Editor Report · Acceptance letter]

12 Nov 2019

PONE-D-19-20568R1 

Improved chemotherapy modeling with RAG-based immune deficient mice 

Dear Dr. WUNDERLICH:

I am pleased to inform you that your manuscript has been deemed suitable for publication in PLOS ONE. Congratulations! Your manuscript is now with our production department. 

With kind regards,

on behalf of

Dr. Daniel Thomas 

Academic Editor

PLOS ONE